# A Holistic Systems Security Approach Featuring Thin Secure Elements for Resilient IoT Deployments

**DOI:** 10.3390/s20185252

**Published:** 2020-09-14

**Authors:** Soodamani Ramalingam, Hock Gan, Gregory Epiphaniou, Emilio Mistretta

**Affiliations:** 1Centre for Engineering Research, Communications and Intelligent Systems, School of Physics, Engineering and Computer Science, Department of Engineering and Technology, University of Hertfordshire, Hatfield AL10 9AB, UK; h.c.gan@herts.ac.uk (H.G.); e.mistretta3@herts.ac.uk (E.M.); 2Warwick Manufacturing Group (WMG), University of Warwick, Coventry CV4 7AL, UK; gregory.epiphaniou@warwick.ac.uk

**Keywords:** IoT, secure elements, key management, threat analysis, B-SPEKE, STRIDE/DREAD

## Abstract

IoT systems differ from traditional Internet systems in that they are different in scale, footprint, power requirements, cost and security concerns that are often overlooked. IoT systems inherently present different fail-safe capabilities than traditional computing environments while their threat landscapes constantly evolve. Further, IoT devices have limited collective security measures in place. Therefore, there is a need for different approaches in threat assessments to incorporate the interdependencies between different IoT devices. In this paper, we run through the design cycle to provide a security-focused approach to the design of IoT systems using a use case, namely, an intelligent solar-panel project called Daedalus. We utilise STRIDE/DREAD approaches to identify vulnerabilities using a thin secure element that is an embedded, tamper proof microprocessor chip that allows the storage and processing of sensitive data. It benefits from low power demand and small footprint as a crypto processor as well as is compatible with IoT requirements. Subsequently, a key agreement based on an asymmetric cryptographic scheme, namely B-SPEKE was used to validate and authenticate the source. We find that end-to-end and independent stand-alone procedures used for validation and encryption of the source data originating from the solar panel are cost-effective in that the validation is carried out once and not several times in the chain as is often the case. The threat model proved useful not so much as a panacea for all threats but provided the framework for the consideration of known threats, and therefore appropriate mitigation plans to be deployed.

## 1. Introduction

Recent developments in sensor networks and the need for smart meters, automated home devices and smart cities have led to the rise in the development of IoT technology-enabling devices to be connected and communicating with each other [1]. Communication between the IoT devices and smart hub is usually wireless and is connected to a network and the Internet via a traditional wireless router. Given the heterogeneous architecture of IoT devices, an adversary might capture the traffic among the different components of an IoT based smart home infrastructure, which relates to passive or active eavesdropping. As another example, an IoT device having insufficient authentication to ascertain identity as in the case of smartphones allows a threat actor to spoof themselves as the owners (impersonation). Lack of authentication allows the takeover of devices for malpractices. Refrigerators, for example, have been shown to send spam emails [2]. It is also easy to launch a Denial of Service (DoS) and Distributed Denial of Service (DDoS) by flooding the hub or router with numerous requests to it by simply knowing and using its IP address. Additionally, these IoT devices come with lightweight software that needs over the air updates, which makes it more vulnerable to malware which can be intercepted more readily than over the wire due to the pervasiveness of the media. These threats have an impact on confidentiality, privacy, data integrity, availability, and access.

Several existing technologies re-deployed in new purpose IoT networks such as Bluetooth, ZigBee, and Wi-Fi are known to increase security vulnerabilities. This is due to the fact that these technologies were not designed for IoT cyberinfrastructures or cyber-physical systems in Industrial IoT deployments [1]. IoT applications and devices make security testing and patching complicated and often expensive due to the increased interdependencies between these devices and applications. They have never incorporated security-by-design principles during their development lifecycles, and security controls are often seen as a top-up. Also, the existing legal and regulatory compliance space for IoT is somewhat fragmented and scattered in terms of the unified approaches for the design, development and testing of these devices and the resilience of their software and firmware. There are significant gaps and overlaps in the existing regulations and standards with regards to appropriate security descriptors to be used when characterising threat landscapes in these environments [3].

Software-based security systems have proven to be vulnerable to attacks even for high-end security applications. Trusted and tamper-proof security platforms cannot be implemented adequately using software-based solutions alone as they have too many entry points that potentially increase the attack surface. If we resolve the problem using specialised operating systems, the application solution is inflexible because the restrictions do not allow the exploitation of the full functionalities of a generic operating system. The issues described above indicate a need to review the way security is holistically implemented for an IoT system [4,5,6,7,8].

The rest of the Section considers a literature review of IoT security focussing on middleware solutions that leads on to the general design strategy adopted in this work for modelling system security using a thin secure element.

### 1.1. Literaure Review and Proposed System Security Model Using Thin Secure Element

Recently, more hardware-assisted techniques have shown potentials to provide a system-wide security protection for IoT devices. The current literature review has emphasised the need to develop and design appropriate security mechanisms with high efficiency and low overhead for lightweight IoT applications deploying hardware architectures [2,9]. Traditionally, such devices use cryptographic methods for handling security aspects of authenticity, message integrity, privacy, and non-repudiation. However, these will only work if these security measures themselves are secure [10]. By using hardware techniques to implement these security measures, any exposure can be encapsulated at vulnerable entry points. The capability of these hardware-based security techniques to offer scalable and resourceful operations under heavy load on microcontrollers, smart cards, and mobile devices is also an area of scientific enquiry [11,12].

A recent survey (2017) survey of various challenges in IoT security [13] provides a standardised taxonomy that helps perform an in depth security analysis including middleware based IoT security. This they achieve by building an abstract model that is composed of interacting elements of the IoT system including humans. The interactions depict the security concerns. All IoT systems can be decomposed to an instance of the model. As a result, it can be used to identify a roadmap for research challenges into IoT security. This roadmap suggests “… much research work is being devoted to developing efficient, robust and low-consumption cryptography for tiny embedded computing and secure protocols for low-power lossy networks. It is essential to adapt and/or design related and equally important sub-systems, such as key management, authentication mechanisms, credential management, and so on…” which is in line with the proposed work in this paper.

Similar to the approach in [13], Xue et al. [14] propose a mathematical modeling framework that captures IoT characteristics with random hypergraphs that have nodes encoding the IoT entities and their interactions at different spatial and temporal dimensionalities. Examples of IoT include RF ID tags, sensors/actuators, end users up to clusters or data centers. The nodes and their interactions are defined by multivalued time-dependent attributes for insights into both its deterministic and stochastic analysis. Such a model is used to identify a list of fundamental research challenges in sensing, the computing paradigm, robustness, energy efficiency and hardware security.

A system that considers a middleware architecture for IoT environments that primarily targets constrained devices such as low RF ID tags and wireless sensor networks is described in [15]. It combines fog computing and cloud paradigm as the middleware to resolve some of the IoT security challenges with respect to Confidentiality, Integrity and Availability (CIA) [16]. This approach enables an efficient use of cloud and server resources by reducing the communication burden on the network and data center on the cloud. The fog layer acts as a gateway to preprocess data at the edge of the network. The middleware sits between devices and applications to act as a medium for communication among devices with different interfaces, architectures and operating systems. The work presents an architectural paradigm that is yet to be tested on a real-world use case. A key difference between this work and our proposed research is that this very architecture has been implemented for a real-world use case.

Research work by Pascal et al. [17,18] consider privacy preserving IoT middleware using Intel’s extended CPU instruction set, Software Guard Extensions (SGX). The SGX allows for the creation of a protected memory region called an enclave where the private keys are stored, and their cryptographic operations are executed. The keys never leave the enclave and are not exposed to the application’s working memory. The increased security of the system comes at the price of reduced performance as indicated by their simulated experiments. The work was designed for desktop and server platforms. In our case, it is designed for smart card applications with a small footprint. The performance of our system meets our requirements as indicated in Section 3.3.

In this paper, we propose the use of a secure cryptoprocessor that some vendors offer as a secure element (SE) [19,20,21,22]. A specific SE used is the Multos™ co-processor P19 to enable the implementation of the most widely used security algorithms and protocols on low power, IoT devices [10]. The Multos tamper-proof hardware prevents manipulation of circuitry and access to the secure memory by physical access [10]. The Multos co-processor is a specially designed chip that has very few avenues of accessibility. The chip contains a specialised environment, and the underlying operating system keeps security as a major requirement. This includes the Multos co-processor being based on a custom-built operating system whose architecture is unknown externally which to a certain extent, makes architectural weaknesses less readily available. More importantly, it is designed as a seamless secure entity which encapsulates the operating system and all other components of the platform such as the bootstrap loader. We exploit this capability of the Multos co-processor as a chip to generate cryptographic keys that are embedded and thereby secure. We design an IoT device to have a strong association with the chip. This will provide an identity through a public-private key pair unique to the chip associated with each IoT device to enable future verification, authorisation, and private communication (channel authentication is handled independently by microcontrollers) between the IoT device and its clients. Therefore, it does not permit the use of a generic toolkit to break into the chip unlike the case of the Linux™ or Windows™ operating systems.

A challenge for IoT security has been remote access. A remote device can be generating its own key-pairs. When the chip enables an IoT device to create and store its own keys in a remote location, the question arises as to how the device controller will receive the public key securely. The traditional way is often to use a third-party trusted authority, which guarantees the origin of a key. However, this does not solve the issue of a spoofing device pretending to be the genuine owner of the generated public key. Our proposed answer is the use of a password authenticated key exchange (B-SPEKE) protocol. B-SPEKE is a variant of a Simple Password Exponential Key Exchange Protocol (SPEKE). The principle behind the B-SPEKE protocol is that the knowledge of the password can be proved without revealing the password. Jablon [23,24,25] describes the method as one that provides “a zero-knowledge password proof (ZKPP) and authenticate-session keys over an unprotected channel, with minimal dependency on infrastructure and proper user behaviour”. More generally, it is a Diffie-Hellman key-exchange [26] where the generator is the hash of a password. Note the distinction between a password used to verify the public key and the keys generated for the signing algorithms. The B-SPEKE password is generated at the time of manufacture whilst the keys are generated on demand for signing the data.

In the general scheme of things, the applicability of our proposed system resides in Layer 1(the Edge layer) of the Cisco™ reference model [27] where IoT is modelled as a combination of wireless sensor networks (WSNs) and cloud services (Figure 1). This is the combination of the first three layers of the reference model: (a) Physical Devices and (b) Controllers, Connectivity and (c) Edge Computing. We propose the introduction of a middleware that has a component attached to physical devices that form a security association with controllers and the servers in the Edge Computing layer.

Several configurations (Figure 2, Figure 3 and Figure 4) are possible within the Edge side layer depending upon whether the chip is used by the device or the controller or both respectively. Figure 2 shows a basic diagram of how the proposed system interfaces with users of the system. The proposed system along with the platform client forms a “thin secure element” (in terms of footprint and power requirement) which interacts with other components of the IoT system as per IoT reference architecture in Figure 1. The platform client can be built along with the security functions as an assembly. Figure 3 shows a configuration of the edge layer where the proposed system is used in an edge device or sensor forming the “thing” of the IoT. The platform client (or API Client) is the device function, which uses the security API (or interface to the security functions) to effect secure communications (such as encryption and data signing) with the controller. The controller, in this case, has enough processing resources (in terms of processing power and memory) to use standard libraries typically provided by off-the-shelf platforms. The Security API provides an option where key generation can be embedded within the proposed system with the association of public keys securely embedded in either the controller or the data processing function or both obviating any vulnerabilities of transit security relationships. The data processing function corresponds to layer three and above of the Cisco IoT reference architecture. Internet access allows secure cloud communications [10] as well as connection to peer-to-peer networks. Figure 4 shows a configuration of the edge layer where the proposed system is used in both an edge device and a controller. In this case, the controller is also one with a small footprint and power requirements requiring secure communications with a data processing function. The option exists to provide separate security associations between the data processing function and the controller as well as the devices, ensuring full traceability between components generating trusted data.

Having considered a bottom-up view where we proposed a solution for a fundamental building block for security, we next approach a holistic view where we consider the framework of design principles for a system focussing on addressing security issues.

### 1.2. Design Model for Security

As previously mentioned, one of the issues with IoT security is that it is often an afterthought. There is a need to include in the design process, procedures where security can be embedded into the system design cycles. In the literature, there exists a variety of threat models that previous developers have used for IT systems [28]. One that was proposed by Microsoft™ [26] was chosen to provide a design process for methodically identifying threats and vulnerabilities as well as providing metrics for evaluating improvements from implementing controls for the identified threats. A more detailed look at what we mean by risk, threats and vulnerabilities and their relationship to each other follows.

A cyber threat refers to an incident that has the potential to harm a system. Intentional threats include spyware, malware, adware companies or malicious actions of disgruntled employees. Worms and viruses are automated threats causing potential harm to systems. Vulnerability refers to a known weakness of an asset that can be exploited successfully by a threat. Examples relate to those involving permissions of people changed or removed at appropriate times, data back-ups, cloud storage, network security, updated licenses of anti-virus software, etc. When a threat exploits a vulnerability, there is a potential for loss or damage defined as the Risk [29]:Risk = ƒ(Threat, Vulnerability)(1)

Equation (1) is a generalisation that Microsoft refines in its threat model. Microsoft classifies threat events using the mnemonic STRIDE (Spoofing, Tampering, Repudiation, Information disclosure-privacy breach or data leak, Denial of service, Elevation of privilege). Each threat in STRIDE is associated with a series of vulnerabilities. Microsoft defined Risk in terms of DREAD [30] as:Risk = (Damage + Reproducibility + Exploitability + Affected Users + Discoverability)/5,(2)
where each parameter is normalised between [0, 10]. The parameters are defined as follows: Damage: measure of the damage to the system, Reproducibility: measure of how reliably the vulnerability can be exploited, Exploitability: difficulty to exploit the vulnerability, Affected Users: number of users affected, Discoverability: measure of ease to discover threat.

DREAD can be visualised as the vulnerability measure for each associated threat. The generic procedure for threat modelling [28] used in this paper is adapted from Microsoft’s security development cycle for software [27] and is listed in Figure 5. The OpenStack Security Group (OSSG) [30,31] has suggested the use of DREAD metric for measuring vulnerability impact in a cloud context. It is acknowledged that scoring techniques using STRIDE for classifying vulnerabilities and DREAD for measuring vulnerability impact are both subjective. Reasoned judgements are to be made while maintaining consistency between the ratings of multiple issues. In this paper, the design cycle listed in Figure 5A has been closely followed. For the following implementation cycle, more importance was given to steps in Figure 5B(d,e) by employing security experts in threat assessment and re-evaluating the hardware configurations.

### 1.3. Contributions

We view the entire design of the IoT platform as a holistic measure with a focus on security. On one hand, we consider the threat modelling of the system, and on the other, the practical improvements that can be made. We adopt a methodology called DREAD/STRIDE used for uncovering security flaws in Software Design Life Cycles (SDLC). In terms of practical considerations, we use a thin secure element and embed the smart circuitry into the device. We apply an on-board asymmetric key-pair generation so that the principle of freshness can be applied to the keys for signing the data. We also implement a zero-knowledge-password-proof (ZKPP) procedure called B_SPEKE to provide public key integrity for users of the public key needed to authenticate the data. Key contributions of this paper can be summarised as follows:Investigation of the state of the art in hardware architectures for developing lightweight IoT security, the notion of security by design and a holistic approach for such security design.We deploy and demonstrate the usefulness of DREAD/STRIDE methodology for uncovering security flaws in SDLC for a real-world use case.Implementing Daedalus—a real-world energy platform with smart solar panels as a use case.Implementing a customized middleware that utilizes a thin secure element for enabling hardware and software security.Designing middleware smart circuitry that is embedded into the IoT device.We implement an on-board asymmetric key-pair generation so that the principle of freshness can be applied to the keys for signing the data.The security procedures for key management also have vulnerabilities, and we address this by implementing a zero-knowledge-password-proof (ZKPP) procedure called B_SPEKE to provide public key integrity for users of the public key needed to authenticate the data.We approach a service architecture by implementing a security API for accessing security functions.We implement a secure cloud computing service (REGUS) to hold data from all the IoT devices. REGUS forms the backbone to the computational process. It establishes a unique security mechanism through chip identity and timestamps usage, demonstrating anti-tampering and authentication with REGUS operations.

### 1.4. Paper Organisation

The remaining paper is organised as follows: In Section 2, we describe a use case called Daedalus, involving an IoT system that monitors a group of solar panels. In Section 3, we apply a systematic threat modelling and evaluation to the existing design to highlight security vulnerabilities. Based on the threat modelling for Daedalus, a recommendation to include a thin secure element is made and implemented. Section 4 considers enhanced security features, namely the management of crypto keys extended to mitigate problems caused by the transport and the refreshment of keys. Section 5 provides the conclusion and future work. We find specific procedures are cost-effective and a model for assessing threats provides a framework for the discussion of risk and vulnerabilities and their resolutions.

## 2. Daedalus: Smart Solar Panels—Use Case

In this section, we consider the functional system design of smart solar panels that are linked as a hierarchy. We consider the process of making a smart solar panel by embedding a Printed Circuit Board (PCB) on to the solar panel that transmits data between the panel and a server.

### Daedalus: Smart Solar Panel System

The focus of Daedalus [32,33] is on renewable energy generated at the point of consumption as rooftop integrated photovoltaics. The enabler for a solar panel to become smart is the addition of a custom-made circuit board as a new IoT device. The circuit board is embedded into each panel for monitoring the power production; this metering data is encoded onboard and then transferred over Wi-Fi to a management unit of the solar panel array. Sensor data measures power generated that is processed as metering data and transferred to the management unit. This process is achieved using ESP8266, which is a low-cost Wi-Fi microchip with full TCP/IP stack and microcontroller capability.

The hardware testbed consists of two distinct modules, the Measurement Module M500 and Management Module M400, as shown in Figure 6. The M500 is attached to each solar panel and the M400 acts as a management module for several M500s in the vicinity. Connected as physically close to each solar panel as possible, the M500 monitors the power production and transfers this metering data. The module contains wireless capabilities along with circuits and components for power measurement. For this application, M500s and M400s both form an internet-enabled system, the M500 being a lightweight edge device and the M400 a more powerful intermediate gateway device. Both modules are embedded in a weatherproof containment. The basis of M500 is illustrated in Figure 7. The voltage and current produced by the panel are monitored and measured with the sensors and the data are then sent to an analog-digital converter (ADC). The converted analog data is sent as digital streams to the ESP8266 micro-controller. The ESP8266 formats the data into MQTT protocol units and forwards that to the Management unit (M400) wirelessly.

## 3. Phase I: System Design for Security

There are several aspects of system design, including function, maintenance, performance, and security. In this section, we focus on the security design aspects. Making changes in one aspect will have an impact on others. This often requires a trade-off between design aspects, which is mostly influenced by business cost. For example, the system architecture for Daedalus can be made extremely secure by fully embedding the control panel shown in Figure 8 excluding the Management Module (M400). However, that would make maintenance almost impossible if we do not have access to the panel or its components.

We start with system design for Daedalus as described in the previous section. We apply a threat model as outlined in Section 1.2 to identify vulnerabilities in detail, provide risk factors associated with each process, and recommend essential security measures. An objective measure for risk is obtained for the entire system using the individual processes that make up the system. Based on such recommendations, the system was re-designed to incorporate strong, on-chip, cryptography that can combat information disclosure, spoofing and tampering as described in Section 1.1 where a ‘secure element’, the Multos P19 chip, was used for this purpose. The design focussed on low and straightforward power protocols such as MQTT with end-to-end cryptography sitting on top, removing complexity, and reducing the attack surface. This was then subjected to the second iteration of threat analysis to ensure lowered system risk. We provided a working prototype of the second iteration and continued with a third iteration (project enhancement) after more funding to improve on the security. Enhancement is typical of commercial projects that have a long-term shelf-life, and the decision to improve on function, maintenance, performance, or security is a business decision based on market demands.

The next subsection describes in detail how we applied the threat model to the Daedalus use-case.

### 3.1. Threat Modelling for Daedalus

In this paper, the assets in Daedalus that we want to secure are (a) data from the solar panels as processed by M500, (b) the transferred data to M400, and (c) data transfer to a central server called REGUS. We identify these assets using a technique introduced in software engineering for highlighting data flow through processes using data flow diagrams (DFDs) [34]. Instead of applying DFD for the maintenance of data from a software engineering perspective, we apply it for the design of security. To this respect, we split threats at vulnerable points of the data into classes using STRIDE. STRIDE provides a basis or checklist for classes of potential types of threats to consider at each vulnerable point. The majority are well known in existing literature [1,35,36](bearing in mind that these tend to grow fast with time). Others may be visualised as part of the procedure. From this respect, the attacker is role-played to generate the threat list. The attacker is assumed to have knowledge and full access to the system and is aware of the vulnerabilities present. Each threat identified is then provided with a risk value, which is calculated in a procedure referred to as DREAD [30]. DREAD provides a categorisation of vulnerabilities to assign risk values based on the threat and the aggregation of those risk values to a single numerical value. In the next subsection, we examine the first step in developing the DFDs.

#### 3.1.1. System Data Composition Based on Communications Diagram (Data Flow Diagrams—DFDs)

We first consider all aspects of Confidentiality, Integrity and Availability (CIA) [16], which are essential for the communication between M400/M500 and between ESP8266 through the A/D converter. The threat analysis is based on communication diagrams only and it focuses on four communicating processes as trust domains extracted from the Daedalus system decomposition shown in Figure 9. Each of these domains is characterised by communication interfaces as follows:TD1—Communication between A/D converter and ESP8266.TD2—Traffic from the Wi-Fi module to the database. The management module, M400 may maintain this.TD4—Communication-related to software update related to ESP8266.TD5—Communication between ESP8266 and management module, M400.TD3—Communication between the ESP8266 and secure element. This is the result of the threat model inclusion and its corresponding DFD is as shown in the Trust Domain and Data Flow Diagram in Section 3.3. This is covered separately in Section 3.2 where the introduction of the thin secure element is proposed.

The data flow decomposition process of Daedalus is modelled using a visual representation of communication paths (VC) within the system and provided in Figure 9. The data flow decomposition diagram (or DFD) shown in Figure 10 consists of data flows (DF), processes (P), data stores (DS) and external entities (EE) as a functional map of the system. The DFD also includes trust domains (TD) which partition the system processes from a security aspect. This enables us to include processes that focus on security concerns rather than software functional flows. The DFD is an important element during the threat identification process that encapsulates the potential threats, if any, to each component of the system. Simplifications have been made in the diagram for this paper, and it chooses to make only four logical functions central to the discussion as the communicating processes (trust domains) for analysis. This will provide an in-depth review of threats in and around these trust domains with a clear taxonomy of those in high priority. The assumption here is that all components within the trust boundaries (attack surfaces) are potential targets and goals for the adversaries. Table 1 provides a detailed list of data flow processes, labels, associated trust domains and attack vectors. As this is a design exercise, which is relatively inexpensive, system components that may not exist are included to have a view on future exposures and impacts to the system. For instance, TD4, which relates to Over the Air (OTA) software updates, was intended as a possible future enhancement.

Having considered where the attack could take place, namely in the four trust domains, the next subsection after a description of the central server (REGUS) considers the type of attacks that can take place.

#### 3.1.2. Renewable Energy Generation Unit Server (REGUS) Server Architecture

The REGUS architecture shown in Figure 11 is composed of a server that provides a service to all the solar panels in the system. The server has access to a relational database via a database management system (SQL 2014) [37]. REGUS is hosted by a web-server (IIS) [38] providing web-based clients access to the REGUS application. This introduces a hierarchy that puts REGUS at the top, followed by M400s and finally the M500s embedded onto the solar panels. There is a concentration factor of M500s to a single M400 and M400s to REGUS. The maximum configuration is determined by the processing performance and memory capacity of each level in the hierarchy. The ESP8266 is not powerful enough to run cryptographic algorithms and lacks secure storage. However, it reads the sensor data and establishes a wireless connection. REGUS was developed as a simple, expandable “Cloud” server with a back-end database to store and process data generated by the solar panels. Communication between the clients and REGUS uses Simple Object Access Protocol (SOAP) [39]. All components use off-the-shelf Microsoft packages. Proprietary software is written in C# using the Windows Communication Foundation (WCF) framework [40,41] to affect communication using SOAP.

Daedalus started with some key requirements including the implementation of smart solar panels, the establishment of trusted data sources with a good measure of security, the investigation of the potential use of block-chains to provide trust, and support of a free-market approach to trading. To support those features, an architecture with a central server was required, which, although could possibly be achieved with an M400, may require resources that were beyond that of the module. M400s are restrained by size and position, but REGUS is not. REGUS can be provisioned with resources that are only limited by cost. In addition, REGUS is not restricted to a physical manifestation. REGUS was conceived very quickly to fulfil the role of that central server. As Daedalus becomes commercialised, the architecture could be re-designed with features moved back to M400 if desired. REGUS can also be realised by cloud computing architectures [42].

The preceding description outlined a typical functional design that had no focus on security aspects. A product based on this design could be delivered, but if security were a concern, the design needs to change. This design process follows incremental design patterns common in many system design life cycles [43] in that appropriate sub-sections of work are put through completion first to de-risk big projects. As the following section shows, an analysis based on security concerns is applied to the existing design to highlight security vulnerabilities.

#### 3.1.3. STRIDE

For Daedalus, the main vulnerability lies between the device and the database. As an example, we focus on threats to the security of the data flowing between the device and the database. This covers trust domains 2 and 5. All the elements of the DFD in the trust domain are systematically examined. All the threats identified are then classified with the STRIDE category. Table 2 shows the threats under the STRIDE category for data received from the Wi-Fi module and data between the Wi-Fi module and the Database. The number of threats identified for each STRIDE category (STRIDE count) for the Wi-Fi module interface (TD2) is respectively; Spoofing(S)—2, Tampering(T)—2, Repudiation(R)—0, Information disclosure(I)—3, Denial of service(D)—1, Elevation of privilege(E)—0. We note that this assessment is based on local expert knowledge of the area and subject to new threat developments. Although there are three counts of Information Disclosure, this is not a major concern within the context of the project because an assumption was made that disclosure of meter-reading data was inconsequential. The analysis so far is an agnostic one, which disregards the consequences of the threats. Once the STRIDE analysis has been completed for all trust domains, we use the DREAD procedure to determine the seriousness of each attack.

#### 3.1.4. DREAD

DREAD is a technique used to establish a ranking for the threats identified. Values are assigned to each DREAD category and an average taken to establish a single risk value for the threat. The assigned values are not predefined and are assigned by informed parties who will assign a value based on the relative weight and priority of the threat. The values should not be treated as absolute measures but rather as guides for improvement. For Daedalus, a security architect evaluated the system and provided the values [44]. Table 3 shows our assignment of Damage, Reproducibility, Exploitability, Affected Users, Discoverability values and the calculation for the average to each database access threat for Daedalus. It is helpful to have a concise description of the threat impact to help with the assignment of the values for the DREAD categories. Once the average is calculated for the trust domain, the view may be further simplified by regarding groups which are colour coded with similar averages when considering priorities for resolution of the risks.

The next stage is to consider the required countermeasures and resolutions for the security risks identified.

#### 3.1.5. Countermeasures: Recommendations for Addressing Risk

This stage is a pivotal stage in the design process because the decision to do additional work rests here. Many of the threats that have been identified in the previous sections would be well documented in the literature, as will their countermeasures. In general, the countermeasures address the security properties of confidentiality, integrity, availability (CIA), freshness and authorisation [45]. Hence, the table that identifies the threat risks for each threat in Table 3 is extended with two columns, namely for recommendations and resolutions, as shown in Table 4.

At this stage, some prototyping (where necessary) can be carried out to help determine the cost and viability of the resolution. It is therefore usual to find that some resolutions may be deferred to a “to-do” list or not regarded as the cost outweigh the perceived risks. Many of the threats will fall into a common category of resolutions. Hence, a table of resolutions (Table 5) is provided, which can be referenced by each threat. Of interest would be design solutions that can address many of the threats. The important point to note is that the process reveals some potential solutions that are beneficial but is not regarded as suitable for immediate implementation and can be recorded for later use.

The recommendations considered are meant for generic desktop systems, which are not compatible with IoT devices dictated by power consumption, footprints, processing capacity, and the like. The equivalent solution suggests the optimisation of the crypto algorithms to fit into an environment suitable for IoT. In the next section, the resolutions for security threats are described. The implementation is an extension to the Daedalus project.

### 3.2. Security Implementation

In this work, we propose a general-purpose cryptographic middleware for the Internet of Things (IoT) devices by extending the Multos P19/P20 processor into a more fully featured, generic security platform. IoT devices are characterised by low power, small footprint of specify dimensions, and low computing capability with the ability to generate and transmit sensitive data. To reduce the computational overheads associated with security aspects, the Multos platform is utilised due to its small footprint in terms of cryptographic overheads. In its original form, the Multos platform allows for a singular planned purpose and custom code written to fulfil that need. This is both a resource-consuming and complicated process, especially as the requirements increase. This work seeks to articulate novel additions to the Multos processor functionalities to make it suitable for IoT. We achieve this by implementing higher-level algorithms, thereby abstracting general security functions. Our proposed system is not restricted to edge devices but also applicable to small factor controllers resulting in a lightweight IoT cryptographic system capable of being integrated to all layers of an IoT network architecture. We provide a mechanism for tamper-proof communication between an IoT device and its controller.

#### 3.2.1. Daedalus Thin Secure Element

The use of a Secure Element (SE) such as the Multos co-processor enables the implementation of the most widely used security algorithms and protocols on low power, IoT like devices. However, the Multos chip cannot be used off the shelf. Multos utilises the same principle as the cards and mobile payment environment. Multos operates in a similar manner. The current Multos platform is sold as a blank chip with low-level cryptographic functions. A developer will typically need to know how to use these functions in security algorithms and protocols to provide the needed security mechanisms for an IoT application. Using an API, we can remove the need to customise the code and the use of the hardcoded option. This reduces significant development time, cost and increases flexibility during updates.

Other companies also offer secure element products similar to Multos meeting the needs of our application [19,20,21,22]. Further, not all products from other companies were fully functional at the time of our implementation. As mentioned in Section 1.1 there are a few configurations for using the Secure Element. Daedalus uses the API client in a controller to interface with the security functions in Figure 4.

#### 3.2.2. On-Boarding Protocols for Multos Functionalities and API

We implement an API to provide access to a set of general-purpose security library functions. Because of the threat analysis carried out in Section 3.1, and based on a survey of popular security functions applicable for IoT devices [36,46,47], we have identified a set of suitable security functions as outlined in Table 6. Based on the threat analysis carried out for Daedalus, a subset of these functions is incorporated as a means of essential and enhanced security.

With the configuration used by Daedalus, communication with the security functions must be affected through a serial port. A communication protocol using Application Protocol Data Units (APDUs) is used as shown in Table 7. The data and commands are sent as request APDUs to the SE, which will then execute the commands and provide a response APDU required by the communication protocol [48]. The design takes the form of two independent entities; one related to the IoT application, the other to the security element communicating via the above said APDUs. The design tailored for the IoT environment in that it uses a primitive form of remote invocation. The security functions are briefly described in Table 8. In the case of Daedalus, to verify the authenticity and integrity of energy-related data from the solar panels, the Hash function using HMAC (Key Hashed Message Authentication Code) [49] was implemented.

#### 3.2.3. REGUS and System Security

REGUS (Section 3.1.2) is used to support the setup of systems providing resources that are not available by the other components (M400 and M500, for instance) of the system. Additional computing resources are used to store and transform the data into information that provide transparency to the users. The support of a more open market approach requires more transparency and accessibility. REGUS provides trustable information to a higher resolution to support transparency. REGUS can provide a more refined control of energy supply, but the infrastructure associated with distribution (e.g., a micro-grid) affects the level of accessibility. Daedalus was conceived to include open market trading. Hence, REGUS was seen to implement this functionality. To achieve an open-market trading model data must be trusted. Some of the properties to provide trusted data are to ensure that relevant data is kept secret and is tamper-proof. In addition, REGUS needs to be protected against malware attacks that cascade impact on different components of the system. In general, such problems already have vast exposure in terms of analyses to provide a secure Information Technology (IT) infrastructure based on the CIA principles [50].

On the client-side of REGUS (Figure 11), a security association is set up between REGUS and M500 which is associated with a solar panel. REGUS holds the HMAC private key for M500, which allows REGUS to authenticate the solar energy readings of the panel collected by M500. In addition, the use of the WCF framework by both M400 and REGUS allows for the configuration of standard security functions for communication made available in the framework [41]. This is reflected in the architectural configuration shown in Figure 12. The M500 employs the services of the security API to provide the HMAC function, but REGUS uses standard libraries.

### 3.3. Phase I: Results and Discussions

This section consists of results obtained by a Client-side test stub that accesses the REGUS application software during development of the REGUS code. REGUS stores time-stamped energy readings from the solar panel in the database shown in Figure 13 using the Chip identity and the Timestamp as primary keys. The data can be retrieved at any time to provide transparency. The Chip identity and Timestamp form a unique pair to access the data. The chip identity (Chip) and the private key are embedded in the chip and can be associated with an owner’s identity. The Timestamp uses a format, which provides a 100 ns count since year 1 [51]. When the data is received from the solar panel, it first must be validated. For the Daedalus project, the data packet is signed using a private key unique to the solar panel. REGUS also must know the key to validate the data. This can be obviated using a public key process as proposed. In addition, the data could be chained as in a blockchain process by transmitting a signature of the previous data and its previous key to provide a stronger form of tamper-proofing.

The result in Figure 14 shows the validation for data (Data) with a transmitted signature (SIGN) and the signature (GEN-SIGN) generated by REGUS from implicit knowledge of the private key. The result is TRUE when both signatures match and FALSE when they do not. Note that if the previous signature were to be used, any error of the data due to transmission, loss, or source problems would cause an unexpected data packet to be validated and the check will fail. Unless there are mechanisms to deal with such failures, there is no way to distinguish between tampering and legitimate reliability issues. The proposal, in this case, is that REGUS can reconstitute the chain after being satisfied that the problem has not been caused by tampering. If REGUS were used to reconstitute the chain, it would introduce REGUS as another point of trust, which may be inevitable anyway as more have to be done to the data to provide usable transactions.

The data that was sent to REGUS is usually compressed as a data packet which is not human readable. Validation is hence not only that the data comes from a source, but the decompressed data itself matches the stored data. Figure 14 shows the interpreted data from compressed data (DATA). Notice that the amps, watts, and energy readings are negative. They should be zero but are the result of imprecision between analogue and digital conversion. This audit trail acts as a useful measure of trust relating to the data at the point of source.

In terms of the thin secure element, a footprint of less than 25 square mm is expected, MULTOS operated between 1.62 V to 5.5 V and at most was using 10 mA to give an idea of power consumption. In terms of computational complexity, we were able to demonstrate a throughput of 180 bytes/s using a cryptographic function such as RSA (Table 8) for digital signatures.

Using the threat modeling outlined in Section 3.1, we propose an end-to-end security design with the following features:i.the use of asymmetric keys for independent validation,ii.unpredictable raw data patterns that add to the quality of freshness,iii.and validation artefacts such as hashed pointers for the data assets using blockchain principles.

The secure chaining of the data assets such as the energy readings, suggests independent protection of data which can be stored in the database, replicated for backups, or distributed for reliability. The chain indicates data can be independently validated from public keys wherever a copy exists provided the public key is trusted. Recent literature suggests the necessity for trustable data assets required for peer-to-peer (P2P) energy trading. The secure chaining of data assets we proposed here provide the trustable assets, also referred to as blockchain oracles [52].

Based on the threat modelling for Daedalus, a recommendation for addressing risk involved the inclusion of the thin secure element. As a result, the Data Flow Diagram (DFD) needed to be updated to allow for a re-evaluation of the risks. The update (See Section 3.1.1) consists of Trust Domain 3 (TD3) as shown in Figure 15 and Figure 16. Furthermore, after the introduction of the secure element, the network functionality of the IoT system in terms of interoperability, resource constraints, scalability and the like must be re-validated. Given the size and complexity of Daedalus, identifying the critical mission components requires a rigorous analytical approach. We defined the concepts of operation (CONOPS) [53] based on the end-to-end flows to establish critical system processes but equally identify critical mission systems in Daedalus related to these processes. We treat this process as a repetitive one as part of building a certain level of protection within Daedalus. We argue that advanced threat actors seek to establish a persistent connection in both hardware and software of our system with full access to the platform and unbounded computational complexity. In this work, we use a rather static approach in the threat modelling process without incorporating elements such as the predicted impact of cyber asset failures. This would typically require a dependency map and a more rigorous Cyber mission impact assessment to introduce resilience [54]. Given the scope of this security analysis and the limited components examined as part of the deliverables, we refrain this analysis in future publications where the maturity level of Daedalus is further established.

## 4. Phase II: Enhancement of Security Measures

The introduction of security measures on the Daedalus project brought the realisation that the additional data generated by the security procedures are themselves prone to vulnerabilities. In particular, the management of crypto keys became prominent in IoT systems that have a small footprint. The life cycle of an IoT system [27] consists of phases such as the manufacturing phase followed by the installation and commissioning phase, and the operational and maintenance phases. An essential part of the management of crypto keys takes part either in the manufacturing or installation phase where the crypto keys are transferred and stored in the required nodes. In the case of Daedalus, a manual offline key distribution mechanism was used with a view to the cost-effective measures that would be important for cheap high-volume systems. The other phase when keys must be managed is during the operation or maintenance phase where keys may be changed for freshness, a concept borrowed from the authentication space [55] or during a breach. As an enhancement of security measures, the API was extended to implement the B-SPEKE protocol to mitigate problems caused by the transport of keys and the refreshment of keys.

### 4.1. Enhanced API

Based on the recommendations for addressing the risks identified by the DREAD analysis, we focused on three security mechanisms with which to enhance the API, namely, secure hashing (SHA256), public key asymmetric signatures (RSA-SSA, [56], ECDSA, [57]) and secure key management (SPEKE, [58]). These mechanisms are supported by encoding functions, key derivation functions (KDF, [59]) and HMAC.

### 4.2. Transport of Keys

One of the problems for the HMAC signatures used in Daedalus was the use of shared keys which meant that the secret keys had to be transported to the source and the destination using a path that is assume secured. The use of asymmetric key procedures obviated the need for that. Consequently, the asymmetric signature algorithms RSA-SSA and ECDSA were implemented in the API. The API simplifies the entire process down to four commands (Table 9) as follows:

The input parameters for RSA and ECDSA differ slightly due to their inherent properties. The choice of the algorithm is developer dependent, and there are specific trade-offs that will need to be considered [60,61].

### 4.3. The Integrity of Public Keys

To keep the design flexible and provide ease of use, the keys are generated on-demand on the secure element, independent of the manufacturer’s intervention, and the secret keys never leave the device’s secure memory. The on-demand nature of the implementation includes a request of key-pair change using the security API (Section 4.2) or automatic key-pair regeneration periodically. When the chip enables an IoT device to generate and stores its own keys in a remote location, the question arises as to how the ecosystem in which the device resides will receive them securely. Sending plain key data is not an option, even though public keys do not need to be hidden, the owner of the key does need to be assured; an un-signed key could have originated from a malicious actor. This is a challenge for IoT security. The traditional way is often to use a third-party trusted authority, which guarantees the origin of a key. However, this does not solve the issue of a remote device generating a key in the wild. Generally, the keys and certificates are generated before deployment. A solution is the use of a password authenticated key exchange (PAKE) protocol [62,63,64]. The principle behind PAKE protocol is that the knowledge of the password can be proved without revealing the password. There is a distinction between a password burnt on the chip at the time of manufacture that is used in the PAKE protocol and the keys generated for the signing algorithms. The PAKE password is generated at the time of manufacture whilst the keys are generated on demand for signing the data. Some of the available algorithms for the PAKE protocol are provided in Table 10 We chose B-SPEKE due to smaller exponential and simple algorithm.

## 5. Conclusions and Further Work

IoT security was lacking even as recently as 2017 and progress has been made but not enough. Key to IoT security is the identity of the device. We show procedures for a system design process focused on security in the way of a threat modelling procedure. In following the process, we found the need to establish security functions compatible with the requirements of IoT devices and arrived at the idea of a thin secure element. With the thin secure element, we provided functions that enable the independent generation of secure identity by themselves. Results show that the threat modelling process is not all-encompassing being that it is not a “silver bullet” solution but provides a process for a holistic, systematic review of threats. To this respect, we do not purport a system that is secure under all circumstances but rather one that satisfies a business and engineering objective in terms of development cost and a reviewed quality target. Our implementation strategy rests on the provision of transparent, auditable data to the IoT devices, and we demonstrated an end-to-end solution.

Every project is exposed to the constraints of a cost-time-scope project management triangle where any one of those three parameters determine the bounds of the project. The implementation of a security system can appear boundless and in conjunction with reliability require an engineering bound to be established. We have noted the limitations of cost and time balance required in the implementation of any security system which puts a limit on the scope of the system but were satisfied that a systematic procedure allows outstanding flaws to be recognised. Vigilance needs to be kept when following the procedure as it is easy to overlook layers of complexity. For instance, the maintenance of the system required a clock-server for time-stamped data assets and the process missed out the identification of clock-server vulnerabilities in the flow. Each data flow hid the several layers that were presented for each flow.

The thin secure element demonstrates a tamper-proof hardware encapsulation. The boundaries of its protection only encompass security functions. Further work would include extending the boundaries to include other functions.

Other areas of further work include key management. Although our enhancement to the system focused on key management, we realised that for large-scaled volume devices, it remains to be an issue. The problem of damage due to Denial of Service (DoS) is certainly beyond the scope of this paper. DoS causes depletion of sensory battery power and reduces efficiency. We also started to develop an interest in general identity theft where machines could be made vulnerable to exploitation. To that respect, we started work on having data that could be independently validated using distributed ledger methods.

## Figures and Tables

**Figure 1 sensors-20-05252-f001:**
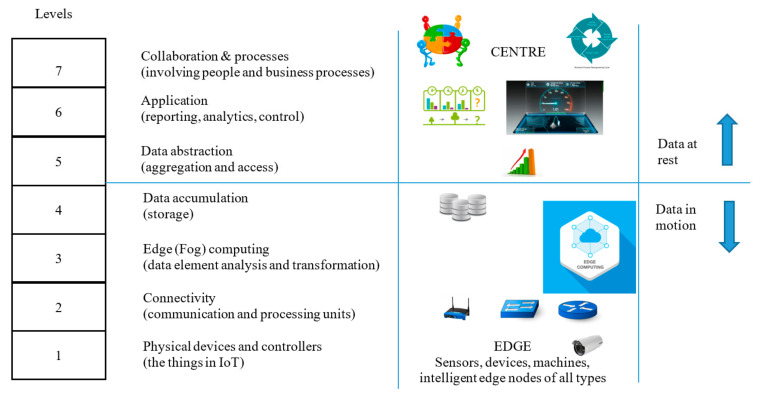
IoT reference architecture.

**Figure 2 sensors-20-05252-f002:**
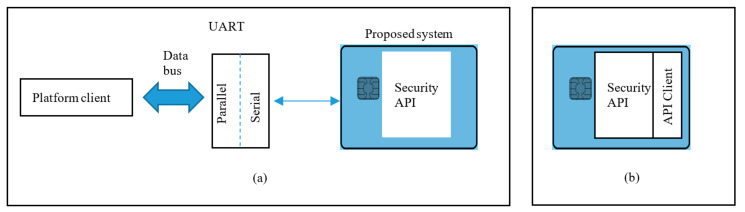
The proposed system consists of a smart card operating system platform integrated with an API to provide security functions whose inputs and outputs are accessible by the transmit and receive lines respectively of a serial port. (**a**) denotes physical interface between an IoT device and the proposed system and (**b**) Configuration 1: alternative solution with a software (instead of a serial port) interface.

**Figure 3 sensors-20-05252-f003:**
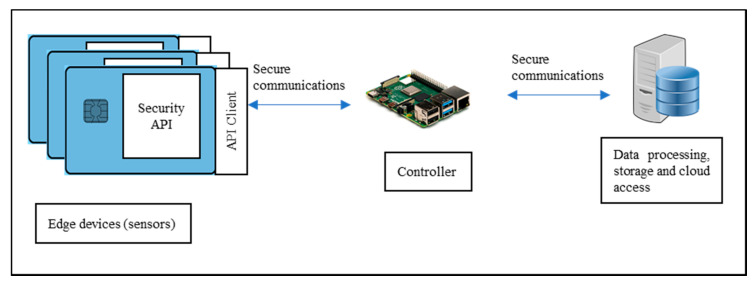
Configuration 2: “Thin secure element” as edge device/sensor. The Controller has enough processing resources to utilise standard security libraries to provide secure communications to the data processing server. The API Client is an external implementation (a micro-controller using a serial port) on the same board as the thin secure element.

**Figure 4 sensors-20-05252-f004:**
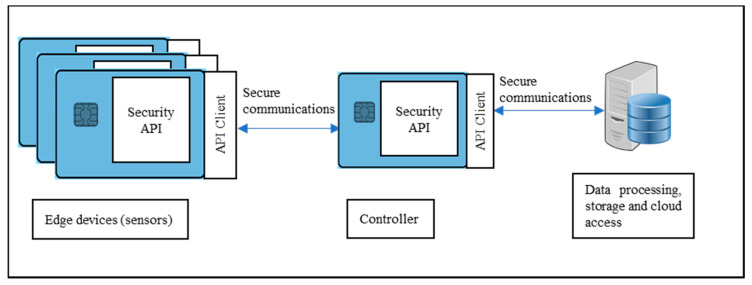
Configuration 3: “Thin secure element” is used in both edge devices and controller. In this case, the controller lacks the processing resources to utilise standard security libraries to provide secure communications.

**Figure 5 sensors-20-05252-f005:**
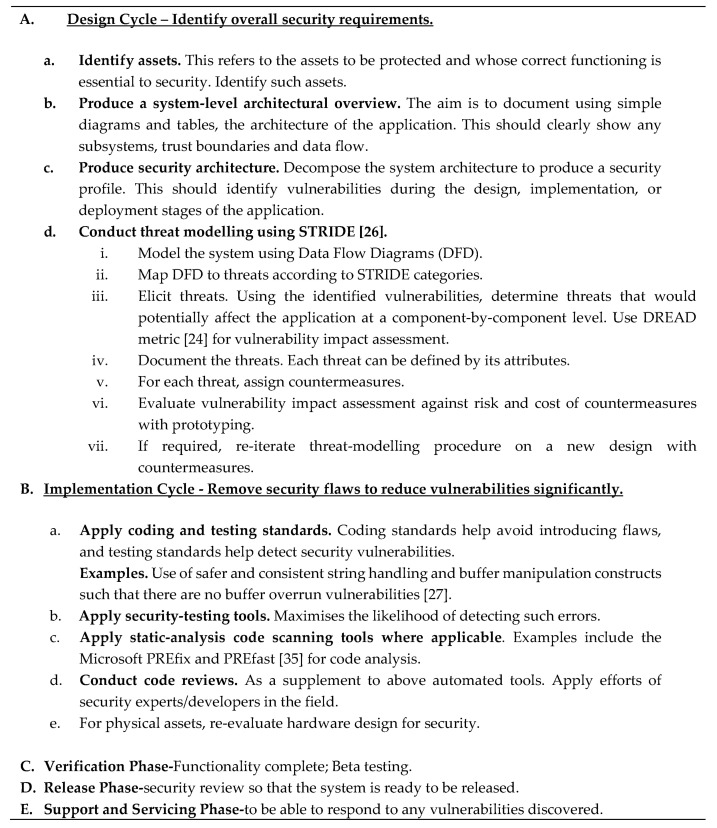
Threat Modelling as part of Security Development Cycle.

**Figure 6 sensors-20-05252-f006:**
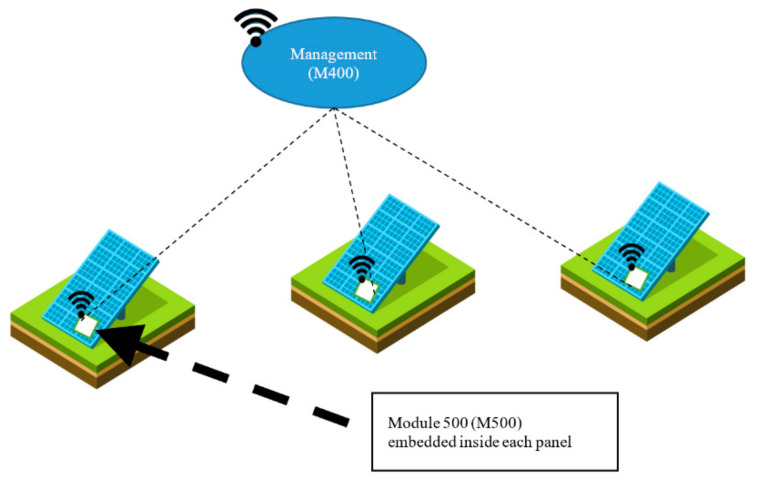
Daedalus: Distributed energy resource asset management system.

**Figure 7 sensors-20-05252-f007:**
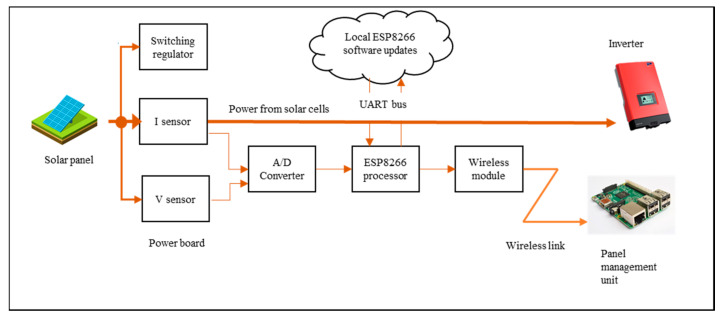
M500 component level architecture.

**Figure 8 sensors-20-05252-f008:**
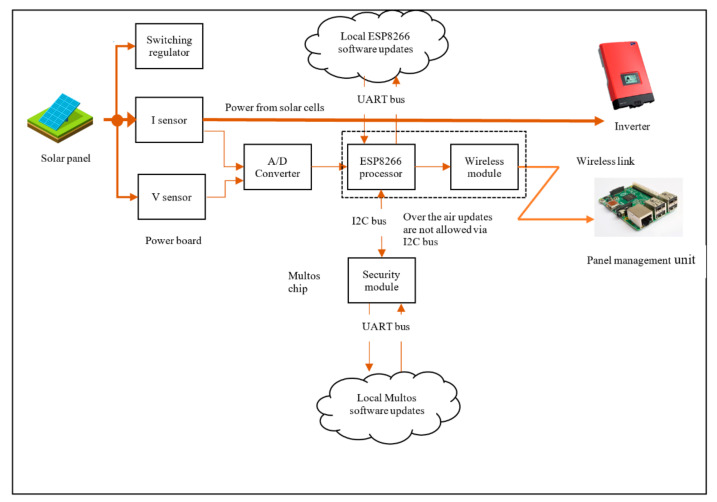
System level architecture showing the communication process.

**Figure 9 sensors-20-05252-f009:**
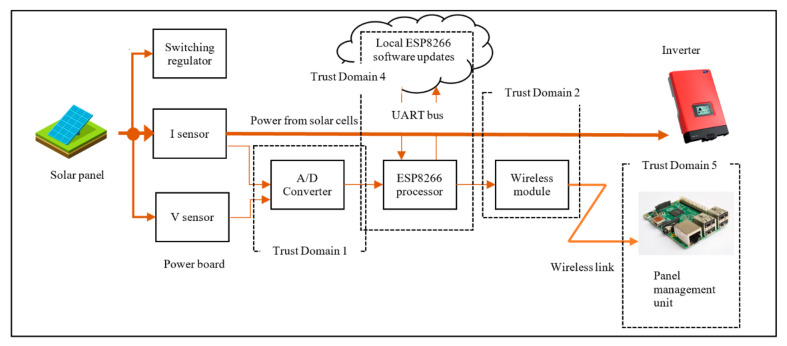
Daedalus Trust Domains without the thin secure element.

**Figure 10 sensors-20-05252-f010:**
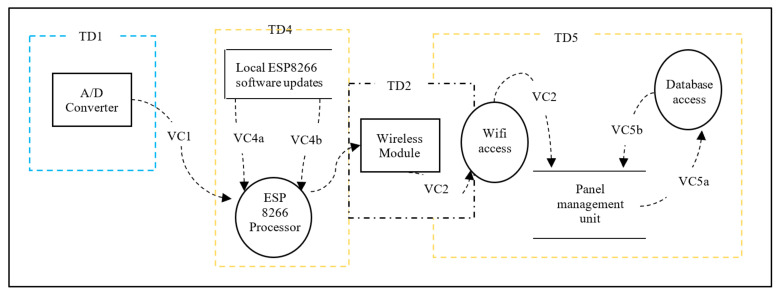
Daedalus Data Flow Diagram for the system without the thin security element.

**Figure 11 sensors-20-05252-f011:**
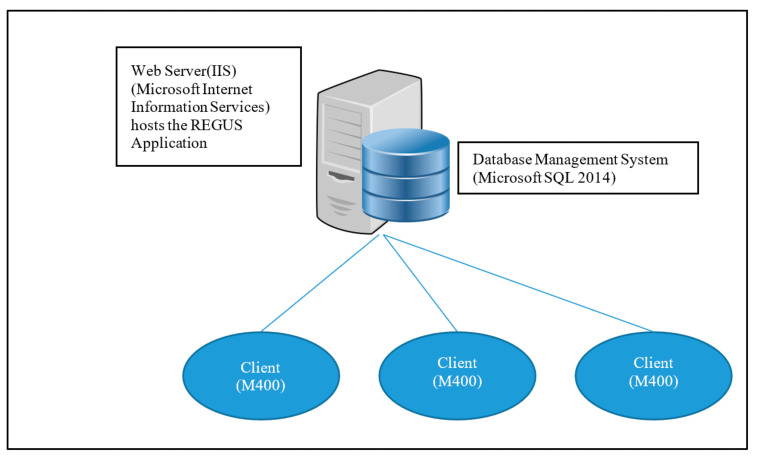
Simple Object Access Protocol (SOAP) was used in Daedalus for client—server communication.

**Figure 12 sensors-20-05252-f012:**
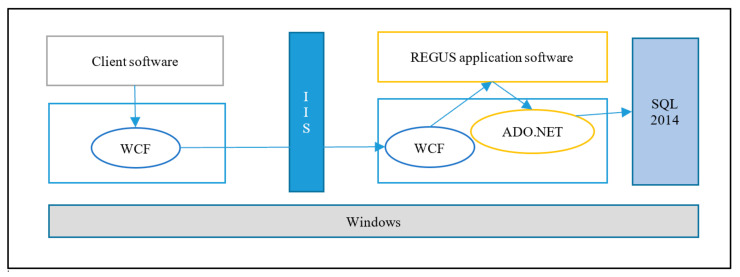
REGUS use of WCF platform for service communication and ADO.NET for database access.

**Figure 13 sensors-20-05252-f013:**
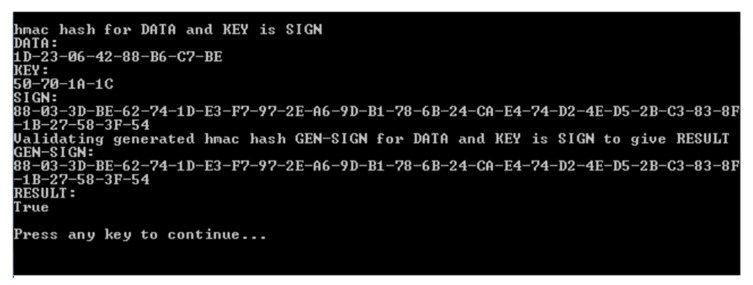
REGUS validating data from the solar panel for a specific data packet.

**Figure 14 sensors-20-05252-f014:**
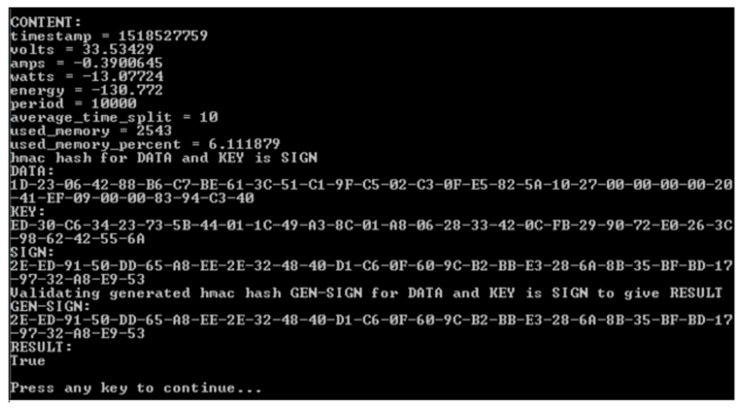
REGUS validating the human readable data is the same as the compressed data received from the solar panel.

**Figure 15 sensors-20-05252-f015:**
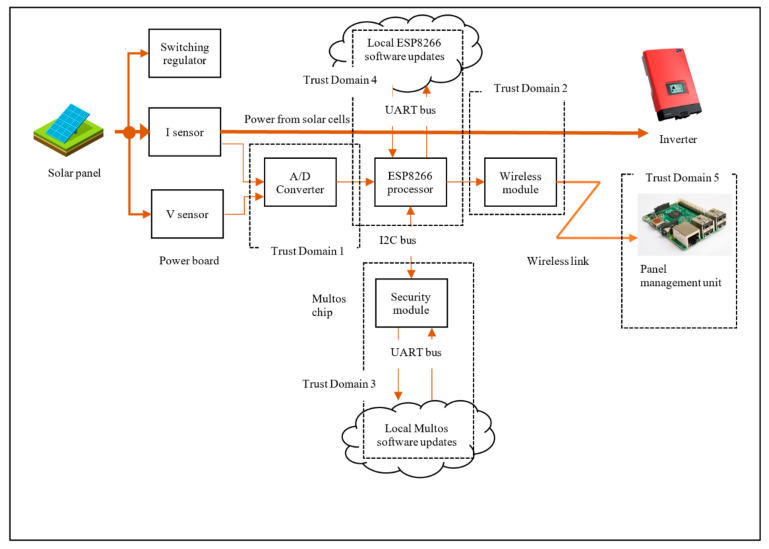
Daedalus Trust Domains with the thin secure element.

**Figure 16 sensors-20-05252-f016:**
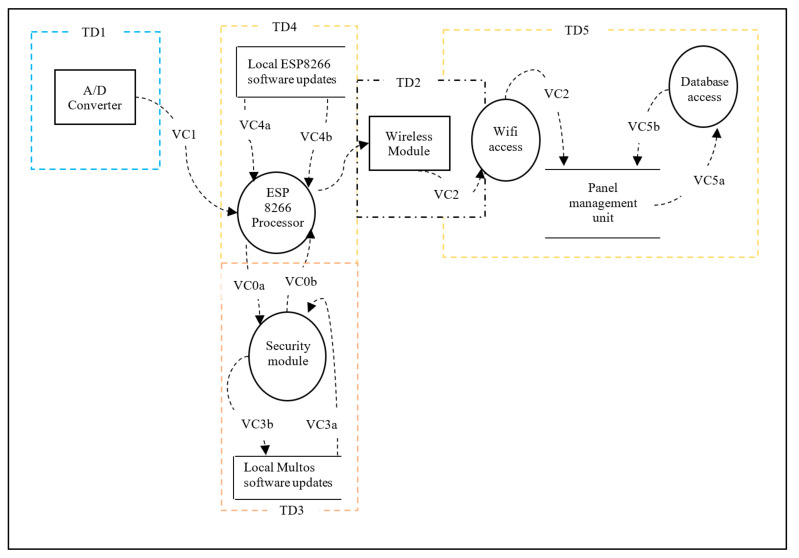
Daedalus Data Flow Diagram with security module.

**Table 1 sensors-20-05252-t001:** Process data flow and threat entries for Daedalus system.

DF ID	Data Process	DF Label	DF Description	Trust Domain	Threat Entry/Exit
1	Database Access	VC2	Data transmission (input)	TD2 & TD5	Wi-fi Module
2		VC5a	Data processing (input)	TD5	SQL Database
3		VC5b	Data processing (output)	TD5	SQL Database
4	ESP 8266 Processor	VC1	Data from A/D converter	TD1	ESP 8266 Processor
5		VC4a	Local software update reception	TD4	ESP 8266 Processor
6		VC4b	Local software update probe	TD4	ESP 8266 Processor & Local ESP8266 Software updates
7		VC0a	Output to security module	TD4 & TD3	ESP 8266 Processor & Security Module & Local ESP8266 Software updates
8		VC0b	Input from security module	TD4 & TD3	ESP 8266 Processor & Security Module
10		VC3b	Input to Multos software update	TD3	Security module & local Multos software update

**Table 2 sensors-20-05252-t002:** Identification of STRIDE threats for Database access in Daedalus.

DF ID	Trust Domain	DF Label	Threat Entry/Exit	STRIDE Count	Threat ID	Threat Event
1	TD2 & TD5	VC2	Wi-fi Module	S = 1	1	Attempting 802.11 Shared Key Authentication with guessed, vendor default or cracked WEP keys
				S = 2	2	Application login theft
				T = 1	3	802.11 frame injection
				T = 2	4	802.11 data replay
				I = 1	5	Krack WPA2 attack
				I = 2	6	TLS logjam attack
				I = 3	7	MITM in wireless communication
				D = 1	8	Wi-fi jamming
2&3	TD5	VC5a &VC5b	SQL Database	S = 1	9	Unauthorised access through replay attack(s)
				S = 2	10	Network Eavesdropping
				T = 1	11	SQL Injection
				T = 2	12	Unauthorised access
				I = 1	13	Unauthorised access
				I = 2	14	Network Eavesdropping
				I = 3	15	Timing analysis
				I = 4	16	Error analysis
				I = 5	17	Malicious data mining
				D = 1	18	D-DoS, DoS, E-DoS attack(s)
				E = 1	19	SQL Injection
				E = 2	20	Unauthorised access
				E = 3	21	Network Eavesdropping

**Table 3 sensors-20-05252-t003:** Generation of DREAD values for threats of Database access in Daedalus.

STRIDE Count	Threat ID	Threat Event	Impact	DREAD: 1 = Low, 5 = High					
				D	R	E	A	D	Avg Score
DFID = 1									
S = 1	1	Attempting 802.11 Shared Key Authentication with guessed, vendor default or cracked WEP keys	Unauthorised access and/or impersonating a legitimate account	1	2	3	4	4	2.8
S = 2	2	Application login theft	Capturing user credentials	1	3	3	4	5	3.2
T = 1	3	802.11 frame Injection	Crafting packets	1	3	2	4	3	2.6
T = 2	4	802.11 data Replay	Capturing 802.11 data frames for later (modified) replay	2	2	4	4	3	3
I = 1	5	Krack WPA2 attack	3rd party eavesdrop confidential information transmitted	4	1	5	5	5	4
I = 2	6	TLS logjam attack	3rd party eavesdrop confidential information transmitted	3	4	3	4	3	3.4
I = 3	7	MITM in wireless communication	Running traditional man-in-the-middle attack tools on an evil twin AP to intercept TCP sessions	2	4	4	3	4	3.4
D = 1	8	Wi-fi jamming	An adversary interrupts communication (data flow) Transmission can be interrupted or blocked	3	2	4	4	4	3.4
**DFID = 2&3**									
S = 1	9	Unauthorized access through replay attack(s)	Falsification of data	2	3	4	3	3	3
S = 2	10	Network Eavesdropping	Impersonating user accounts, stolen credentials	1	2	3	3	3	2.4
T = 1	11	SQL Injection	Run arbitrary commands in the database, manipulate, erase data	2	5	4	4	4	3.8
T = 2	12	Unauthorized access	Alteration of tables, modification of data	2	3	4	3	4	3.2
I = 1	13	Unauthorized access	Stolen records	1	3	4	4	5	3.4
I = 2	14	Network Eavesdropping	Unauthorized interception of information	1	2	3	3	3	2.4
I = 3	15	Timing analysis	Recovering private entries from private columns	1	2	3	3	3	2.4
I = 4	16	Error analysis	Exception conditionsTarget non-validated user inputWeak dynamic SQL queries	1	4	3	3	3	2.8
I = 5	17	Malicious data mining	Information gatheringSQL injection	1	2	2	3	2	2
D = 1	18	D-DoS, DoS, E-DoS attack(s)	Limit or prohibit access to legitimate usersExecuting non-optimized codeBad resource allocation and management policy	3	4	1	3	4	3
E = 1	19	SQL Injection	Run system commands	2	5	4	4	4	3.8
E = 2	20	Unauthorized access	Unauthorized command execution, table creation, deletion	1	3	4	4	5	3.4
E = 3	21	Network Eavesdropping	Execute arbitrary commandsDatabase alteration, deletion	2	4	3	3	4	3.2

**Table 4 sensors-20-05252-t004:** Recommendations and resolutions for threats of database access.

STRIDE Count	Threat ID	Threat Event	Impact	DREAD	Recommended Action	Resolution ID
DFID = 1						
S = 1	1	Attempting 802.11 Shared Key Authentication with guessed, vendor default or cracked WEP keys	Unauthorised access and or impersonating a legitimate account	2.8	Disable WEP/WPA. Provide 802.11X and investigate options	5, 13
S = 2	2	Application login theft	Capturing user credentials	3.2	Strong encryption, strong passwords, adequate password policy	5, 2, 16
T = 1	3	802.11 frame Injection	Crafting packets	2.6	Consider a Robust Secure Network implementation	5, 13, 18
T = 2	4	802.11 data Replay	Capturing 802.11 data frames for later (modified) replay	3	Use of Kerberos for authentication within IEEE 802.1X	5, 13, 18
I = 1	5	Krack WPA2 attack	3rd party eavesdrop confidential information transmitted	4	Use available counter-measure patches	5, 3, 18
I = 2	6	TLS logjam attack	3rd party eavesdrop confidential information transmitted	3.4	Disable support for export- grade cipher suites, Use ECDHE instead of DHE, Reduce TLS handshake timeout	5, 3, 10
I = 3	7	Man-In-The-Middle in wireless communication	Running traditional man-in-the-middle attack tools on an evil twin Access Point (AP) to intercept TCP sessions	3.4	TLS encryption, RADIUS authentication server, consider mTesla protocol in the given architecture	5, 3
D = 1	8	Wi-fi jamming	An adversary interrupts communication (data flow) Transmission can be interrupted or blocked	3.4	Explore anti-jamming features, difficult to block	13
**DFID = 2&3**						
S = 1	9	Unauthorised access through replay attack(s)	Falsification of data	3	Strong authentication, identity management, key freshness. Use of Windows authentication	5, 2, 1, 15
S = 2	10	Network Eavesdropping	Impersonating user accounts, stolen credentials	2.4	Use authentication based on key exchange. Discovery scanners. Use an access control list. Reject packets originating from outside your local network that claim to originate from within	5
T = 1	11	SQL Injection	Run arbitrary commands in the database, manipulate, erase data	3.8	Input sanitisation	13, 8, 17
T = 2	12	Unauthorised access	Alteration of tables, modification of data	3.2	Strong hashing for tampering detection purposes, timestamps, salting. Use of Windows authentication	1, 6, 17
I = 1	13	Unauthorized access	Stolen records	3.4	Encrypted database systems including transactions. Use of Windows authentication	5, 1, 3, 16
I = 2	14	Network Eavesdropping	Unauthorized interception of information	2.4	SSL, IPSEC	5, 3
I = 3	15	Timing analysis	Recovering private entries from private columns	2.4		5, 3
I = 4	16	Error analysis	Exception conditions Target non-validated user input Weak dynamic SQL queries	2.8	Effective filtering. Trusted connections to the database. Exception handling	5, 3
I = 5	17	Malicious data mining	Information gathering SQL injection	2		5, 3
D = 1	18	D-DoS, DoS, E-DoS attack(s)	Limit or prohibit access to legitimate users Executing non-optimized code Bad resource allocation and management policy	3	No effective countermeasure at the database level	13
E = 1	19	SQL Injection	Run system commands	3.8	Restricted accounts	14, 13
E = 2	20	Unauthorized access	Unauthorized command execution, table creation, deletion	3.4		9, 13
E = 3	21	Network Eavesdropping	Execute arbitrary commands Database alteration, deletion	3.2	Use an access control list	5, 3

**Table 5 sensors-20-05252-t005:** Resolutions for the threat events in DREAD determined during the first phase of Daedalus.

ID	Resolution
1	Windows authentication used on SQL DB (Regus).
2	The broker and clients authenticate their secured IDs.
3	In Daedalus, confidentiality was not a major concern as it related to meter reading.
4	Disable Over the Air firmware updates
5	End-to-end cryptography (signatures) deployed
6	Salting not implemented (out of scope)
7	Clock security has not been dealt with
8	Sanitisation to be considered (out of scope)
9	Standard administrative privileges (ACL) set
10	Not applicable in the present version
11	Considered as a low probability threat based on complexity and cost implications
12	The ‘embedded’ systems approach does not permit external access to memory
13	Not mitigated in this version
14	Account restriction to be considered (out of scope)
15	Integrity of public key to be considered (out of scope)
16	End-to-end Encryption to be considered (out of scope)
17	Means to independently validate data assets considered, possibly blockchain (out of scope)
18	Proposed solution. Part of the end-to-end key management in Section 4

**Table 6 sensors-20-05252-t006:** General software security functions.

Security Feature	Example Algorithms/Protocols	Remarks
Asymmetric (public) Key Encryption	RSA	Allow the simple generation of key pairs, encryption & decryption operations, and provide secure on chip storage of private keys
	ECC	
Symmetric Encryption	AES	Allow the simple generation or input of secret keys, encryption & decryption operations, and provide secure on chip storage for secret keys
	DES	
Cryptographic Hash	SHA/1/2/3	Allow simple generation of hash digests from data
Keyed Cryptographic Hash	HMAC-SHA1/2/3	Allow simple generation of HMAC digests from data and a secret key. Provide secure storage for secret keys
Asymmetric Key Signature	RSA-SSA	Allow the simple generation of key pairs, signing & verification operations, and provide secure on chip storage of private keys
	ECDSA	
	DSA	
Key Exchange	PAKE	Built in PAKE protocol for bootstrapping/on-boarding using unique ID & secret for each secure chip. Allow simple use of DH protocol, securely store session keys.
	DH/ECDH	
Certification	CA root certificates	Have common CA root certificates built into the system. Allow simple verification of certificates signed by different CA’s

**Table 7 sensors-20-05252-t007:** Application Data Protocol definitions for invoking Daedalus security API.

Command APDU		
Field Name	Length (Bytes)	Description
CLA	1	Instruction class - indicates the type of command, e.g., interindustry or proprietary
INS	1	Instruction code - indicates the specific command, e.g., “write data”
P1	1	Instruction parameter for the command
P2	1	Instruction parameter for the command
Lc	0,1 or 3	Encodes the number of bytes of command data as follows:
		0 bytes denotes 0
		1 byte with a value from 1 to 255 denotes the same value
		3 bytes, the first of which must be 0, denote in the range 1 to 65 535 (all three bytes may not be zero)
Data	0 to 65535	Command data
Le	0,1, 2 or 3	Encodes the maximum number of response bytes expected.
		0 bytes denotes 0
		1 byte in the range 1 to 255 denotes that value, or 0 denotes 256
		2 bytes (if 3 byte Lc was present in the command) in the range 1 to 65 535 denotes that value, or two zero bytes denotes 65 536
		3 bytes (if Lc was not present in the command), the first of which must be 0, denote the same way as two-byte Le
**Response APDU**		
Data	0 to 65536	Response data
SW1-SW2	2	Command processing status, e.g., 90 00 (hexadecimal) indicates success

**Table 8 sensors-20-05252-t008:** Overview of available commands in the security API.

Command	Description	Input	Output
HASH	Creates a hash digest of data using the desired algorithm	Desired hash algorithm & Data to hash	Hash digest
RSASSA-GENKEYPAIR	Generates a key pair for use with the RSA-SSA algorithm	Required key size & public exponent	Index to key pair in secure storage
RSASSA-GETPUBKEY	Returns the public key information from an RSA-SSA key pair	Key pair index	Public key data
RSASSA-SIGN	Signs data using the private key from an RSA-SSA key pair	Key pair index, desired hash algorithm, desired encoding scheme & data to sign	Signature
RSASSA-VERIFY	Verifies an RSA-SSA signature	Hash algorithm used, encoding scheme used, Public key information of signer, data signed & signature	True or false indicating if signature is valid
ECDSA-GENKEYPAIR	Generates a key pair for use with the ECDSA algorithm	Desired curve	Index to key pair in secure storage
ECDSA-GETPUBKEY	Returns the public key information from an ECDSA key pair	Key pair index	Public key data
ECDSA-SIGN	Signs data using the private key from an ECDSA key pair	Key pair index, desired hash algorithm & data to sign	Signature
ECDSA-VERIFY	Verifies an ECDSA signature	Curve used, hash algorithm used, public key information of signer, data signed & signature	True or false indicating if signature is valid
BSPEKE-INIT	Performs initial steps of the B-SPEKE protocol	None	ID & calculated client data (A)
BSPEKE-CALC	Calculates the shared secret	Calculated server data (B)	Client secret verification message (M1)
BSPEKE-VERIFY	Verifies the shared secret is correct & derives the secret key from the shared secret	Server secret verification message (M2), desired key length & desired key derivation function	True or false indicating if M2 is valid
BSPEKE-GETKEY	Returns any stored public key signed with a HMAC digest using the generated secret key.	Key pair index, desired HMAC algorithm	Public key data & HMAC digest

**Table 9 sensors-20-05252-t009:** API commands to support the generation of asymmetric signature algorithms.

	Commands		Input-Output Parameters
1.	Generate key pair	a.	Input desired parameters such as key size.
		b.	The output is an index to the key pair.
2.	Retrieve public key	a.	Input is an index to a key pair.
		b.	The output is the public key relating to the index.
3.	Sign data	a.	Input is an index to a key pair and the data to be signed as well as any encoding schemes.
		b.	The output is the signature.
4.	Verify signature	a.	Input would be the parameters from the originator such as their public key, the signature and the data being signed.
		b.	The output would be true or false depending on if it is valid.

**Table 10 sensors-20-05252-t010:** Choice of algorithms for PAKE protocol.

Algorithm	Remarks
Secure Remote Password (SRP) [65]	A patent free augmented PAKE algorithm made at a time when algorithms such as SPEKE (and variants, below) were under patent. SRP requires a large exponentiation, the Multos platform does not support the use of an exponent of the size required.
Simple Password Exponential Key Exchange (SPEKE) [24].	A Simpler algorithm to SRP. Uses smaller exponentiations. SPEKE is a balanced PAKE algorithm, so the secret key must be stored on the server. The server being compromised would allow an attacker to masquerade as a client.
B-SPEKE [23,25]	An augmented version of SPEKE. The server stores a verifier, Loss of the verifier would not allow an attacker to masquerade as a client, unless the discrete logarithm problem was solved.
W-SPEKE IEEE P1363 [36]	Another augmented version of SPEKE. Uses a larger exponentiation like SRP.

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
