# Peer review of "A Holistic Systems Security Approach Featuring Thin Secure Elements for Resilient IoT Deployments"

_sensors, 2020, doi:10.3390/s20185252_

Round 1

Reviewer 1 Report

In this paper, the authors provide a security-focused approach to the design of IoT systems using a use case, which is an intelligent solar-panel project called Daedalus. They utilize STRIDE/DREAD approaches to identify vulnerabilities using a thin secure element that is an embedded, tamper proof microprocessor chip that allows the storage and processing of sensitive data. The manuscript is very well organized and I only have the following suggestions:

  1. What are contributions of the paper? It seems that this paper borrows the existing platforms and approaches such as Daedalus and DREAD. Please provide a subsection summarizing the main contributions.
  2. The related work should be better discussed and contrasted. For example there are important works in formalizing the IoT design problems which can also help with the security aspects. Here are some examples:  Fundamental Challenges Toward Making the IoT a Reachable Reality: A Model-Centric Investigation. ACM Trans. Des. Autom. Electron. Syst. 22, 3, Article 53 (May 2017), 25 pages. DOI:https://doi.org/10.1145/3001934
  1. In experiments, there are no comparisons with some state-of-the-art techniques in the field to validate your approach is better. Also, the reviewer wonders how you can prove your system is secure under all circumstances.
  2. In Section 3.1.3, authors talked about DREAD, which is a technique to establish a ranking for the threads identified. The reviewer wonders how you assign a value to each thread. Is it predefined or somehow calculated? If it is predefined, how to prove the values are correct?

Minor issues:

  1. Error! Reference source not found..

Author Response

We are very grateful to the reviewers for their insightful comments on the paper. We have addressed their feedback as follows:

Reviewer 1

  1. In this paper, the authors provide a security-focused approach to the design of IoT systems using a use case, which is an intelligent solar-panel project called Daedalus. They utilize STRIDE/DREAD approaches to identify vulnerabilities using a thin secure element that is an embedded, tamper proof microprocessor chip that allows the storage and processing of sensitive data. The manuscript is very well organized and I only have the following suggestions:

What are contributions of the paper? It seems that this paper borrows the existing platforms and approaches such as Daedalus and DREAD. Please provide a subsection summarizing the main contributions. 

Action

                Addressed in a new Section 1.3 titled  ‘Contributions’. Details are as follows:

  • Contributions

We view the entire design of the IoT platform as a holistic measure with a focus on security. On one hand we consider the threat modelling of the system, and on the other, the practical improvements that can be made. We adopt a methodology called DREAD/STRIDE used for uncovering security flaws in Software Design Life Cycles (SDLC). In terms of practical considerations, we use a thin secure element and embed the smart circuitry into the device. We apply an on-board asymmetric key-pair generation so that the principle of freshness can be applied to the keys for signing the data. We also implement a zero-knowledge-password-proof (ZKPP) procedure called B_SPEKE to provide public key integrity for users of the public key needed to authenticate the data. Key contributions of this paper can be summarised as follows:

  • Investigation of the state of the art in hardware architectures for developing lightweight IoT security, the notion of security by design and a holistic approach for such security design.
  • We deploy and demonstrate the usefulness of DREAD/STRIDE methodology for uncovering security flaws in Software Design Life Cycles (SDLC) for a real-world use case.
  • Implementing Daedalus - a real-world energy platform with smart solar panels as a use case.
  • Implementing a customized middleware that utilizes a thin secure element for enabling hardware and software security. 
  • Designing middleware smart circuitry that is embedded into the IoT device .
  • We implement an on-board asymmetric key-pair generation so that the principle of freshness can be applied to the keys for signing the data. 
  • The security procedures for key management also have vulnerabilities, and we address this by implementing a zero-knowledge-password-proof (ZKPP) procedure called B_SPEKE to provide public key integrity for users of the public key needed to authenticate the data.
  • We approach a service architecture by implementing a security API for accessing security functions.
  • We implement a secure cloud computing service (REGUS) to hold data from all the IoT devices. REGUS forms the backbone to the computational process. It establishes a unique security mechanism through chip identity and timestamps usage, demonstrating anti-tampering and authentication with REGUS operations.
  1. The related work should be better discussed and contrasted. For example there are important works in formalizing the IoT design problems which can also help with the security aspects. Here are some examples:  Fundamental Challenges Toward Making the IoT a Reachable Reality: A Model-Centric Investigation. ACM Trans. Des. Autom. Electron. Syst. 22, 3, Article 53 (May 2017), 25 pages. DOI:https://doi.org/10.1145/3001934  

Action         

                Section 1.1 which was previously titled ‘Thin Secure Element’ is renamed as ‘Literature Review…’ and new material included to cover the state of the art in related middleware based IoT security.  The paper suggested by the reviewer has also been considered as part of this work. It reads as follows:

  • Literaure Review and Proposed System Security Model using Thin Secure Element

Recently, more hardware-assisted techniques have shown potentials to provide a system-wide security protection for IoT devices. The current literature review has emphasised the need to develop and design appropriate security mechanisms with high efficiency and low overhead for lightweight IoT applications deploying hardware architectures [4, 5]. Traditionally, such devices use cryptographic methods for handling security aspects of authenticity, message integrity, privacy, and non-repudiation. However, these will only work if these security measures themselves are secure [6]. By using hardware techniques to implement these security measures, any exposure can be encapsulated at vulnerable entry points. The capability of these hardware-based security techniques to offer scalable and resourceful operations under heavy load on microcontrollers, smart cards, and mobile devices is also an area of scientific enquiry [7, 8].

A recent survey (2017) of various challenges in IoT security [9] provides a standardised taxonomy that helps perform an in depth security analysis including middleware based IoT security. This they achieve by building an abstract model that is composed of interacting elements of the IoT system including humans.  The interactions depict the security concerns. All IoT systems can be decomposed to an instance of the model.  As a result, it can be used to identify a roadmap for research challenges into IoT security. This roadmap suggests “…much research work is being devoted to developing efficient, robust and low-consumption cryptography for tiny embedded computing and secure protocols for low-power lossy networks. It is essential to adapt and/or design related and equally important sub-systems, such as key management, authentication mechanisms, credential management, and so on…” which is in line with the proposed work in this paper.

Similar to the approach in [9], Xue at al [10] propose a mathematical modeling framework that captures IoT characteristics with random hypergraphs that have nodes encoding the IoT entities and their interactions at different spatial and temporal dimensionalities. The nodes and their interactions are defined by multivalued time-dependent attributes for insights into both its deterministic and stochastic analysis. Examples of IoT include RF ID tags, sensors/actuators, end users up to clusters or data centers. Such a model is used to identify a list a fundamental research challenges in sensing, the computing paradigm, robustness, energy efficiency and hardware security.

A system that considers a middleware architecture for IoT environments that primarily targets constrained devices such as low RF ID tags and wireless sensor networks is described in [11]. It combines fog computing and cloud paradigm as the middleware to resolve some of the IoT security challenges with respect to Confidentiality, Integrity and Availability (CIA) [12].  This approach enables an efficient use of cloud and server resources by reducing the communication burden on the network and data center on the cloud. The fog layer acts as a gateway to preprocess data at the edge of the network. The middleware sits between devices and applications to act as a medium for communication among devices with different interfaces, architectures and operating systems. The work presents an architectural paradigm that is yet to be tested on a real-world use case. A key difference between this work and our proposed research is that this very architecture has been implemented for a real-world use case.

Research work by Pascal et al [13, 14] consider privacy preserving IoT middleware using Intel’s extended CPU instruction set, Software Guard Extensions (SGX). The SGX allows for the creation of a protected memory region called an enclave where the private keys are stored, and their cryptographic operations are executed. The keys never leave the enclave and are not exposed to the application's working memory. The increased security of the system comes at the price of reduced performance as indicated by their simulated experiments. The work was designed for desktop and server platforms. In our case, it is designed for smart card applications with a small footprint. The performance of our system meets our requirements as indicated in Section 3.3.

  1. In experiments, there are no comparisons with some state-of-the-art techniques in the field to validate your approach is better. Also, the reviewer wonders how you can prove your system is secure under all circumstances.

Action: Done. Following highlighted text included in Section 5 on Conclusion, lines 699-701.

Results show that the threat modelling process is not all-encompassing being that it is not a “silver bullet” solution but provides a process for a holistic, systematic review of threats. To this respect, we do not purport a system that is secure under all circumstances but rather one that satisfies a business and engineering objective in terms of development cost and a reviewed quality target. Our implementation strategy rests on the provision of transparent, auditable data to the IoT devices, and we demonstrated an end-to-end solution.

  1. In Section 3.1.3, authors talked about DREAD, which is a technique to establish a ranking for the threads identified. The reviewer wonders how you assign a value to each thread. Is it predefined or somehow calculated? If it is predefined, how to prove the values are correct?

Action: Done. Following highlighted text included in Section 3.1.4 on DREAD, lines 443-445.

DREAD is a technique used to establish a ranking for the threats identified. Values are assigned to each DREAD category and an average taken to establish a single risk value for the threat. The assigned values are not predefined and are assigned by informed parties who will assign a value based on the relative weight and priority of the threat. The values should not be treated as absolute measures but rather as guides for improvement.

  1. Minor issues:

       Error! Reference source not found.. 

Action:  We couldn’t find one such error in the document. No action taken.

Reviewer 2 Report

This is a good work that addresses key challenges in the area of IoT deployments. I would however suggest a more clear comparison with the state-of-the-art, thus highlighting the significance of the approach.

Author Response

This is a good work that addresses key challenges in the area of IoT deployments. I would however suggest a more clear comparison with the state-of-the-art, thus highlighting the significance of the approach. 

Action

Same as Reviewer 1, item 2. Addressed. 

Section 1.1 which was previously titled ‘This Secure Element’ is renamed as ‘Literature Review…’ and new material included to cover the state of the art in related middleware based IoT security. 

Reviewer 3 Report

In this manuscript, the authors have proposed a secure strategy to design of IoT systems in an intelligent solar panel project named Daedalus. For this purpose, they used an embedded thin secure chip that can store and process sensitive data. The paper discusses full details and is well-written. In my opinion, it can be accepted for Sensors. However, addressing the following issues can improve the quality of the work.

1) As energy consumption and processing time are crucial in IoT systems, I recommend discussing the computational complexity and the processing time of the proposed approach.

2) Line 160 (the caption of Figure 3): the phrase "The server." should be removed.

3) Line 320: there is a mistake in the number of subsection. Subsection 2.2 is between 3.1.1 and 3.1.2.

4) Line 496: there is a mistake in the number of subsection 3.2. It should be changed to 3.3.

Author Response

In this manuscript, the authors have proposed a secure strategy to design of IoT systems in an intelligent solar panel project named Daedalus. For this purpose, they used an embedded thin secure chip that can store and process sensitive data. The paper discusses full details and is well-written. In my opinion, it can be accepted for Sensors. However, addressing the following issues can improve the quality of the work. 

  1. As energy consumption and processing time are crucial in IoT systems, I recommend discussing the computational complexity and the processing time of the proposed approach.

Action

Done.  Following  text included in

  • Phase I: Results and Discussions , Lines 589-592.

In terms of the thin secure element, a footprint of less than 25 square mm is expected, MULTOS operated between 1.62v to 5.5V and at most was using 10 mA to give an idea of power consumption. In terms of computational complexity, we were able to demonstrate a throughput of 180bytes/sec using a cryptographic function such as RSA (Table 8) for digital signatures.

  1. Line 160 (the caption of Figure 3): the phrase "The server." should be removed.

Done, phrase deleted.

  1. Line 320: there is a mistake in the number of subsection. Subsection 2.2 is between 3.1.1 and 3.1.2.

Done. Renumbering was carried out.

  1. Line 496: there is a mistake in the number of subsection 3.2. It should be changed to 3.3.

Done. Renumbering was carried out.

Round 2

Reviewer 1 Report

The authors have address most of reviewers concerns.

Author Response

A spell check has been carried out.